# Construction of ceRNA Networks at Different Stages of Somatic Embryogenesis in Garlic

**DOI:** 10.3390/ijms24065311

**Published:** 2023-03-10

**Authors:** Yunhe Bai, Min Liu, Rong Zhou, Fangling Jiang, Ping Li, Mengqian Li, Meng Zhang, Hanyu Wei, Zhen Wu

**Affiliations:** 1College of Horticulture, Nanjing Agricultural University, Nanjing 210095, China; 2Key Laboratory of Horticultural Plant Biology and Germplasm Innovation in East China, Ministry of Agriculture, Nanjing 210095, China; 3Department of Food Science, Aarhus University, Agro Food Park 48, 8200 Aarhus, Denmark

**Keywords:** garlic, somatic embryogenesis, ceRNA network, plant hormone signal

## Abstract

LncRNA (long non-coding RNA) and mRNA form a competitive endogenous RNA (ceRNA) network by competitively binding to common miRNAs. This network regulates various processes of plant growth and development at the post-transcriptional level. Somatic embryogenesis is an effective means of plant virus-free rapid propagation, germplasm conservation, and genetic improvement, which is also a typical process to study the ceRNA regulatory network during cell development. Garlic is a typical asexual reproductive vegetable. Somatic cell culture is an effective means of virus-free rapid propagation in garlic. However, the ceRNA regulatory network of somatic embryogenesis remains unclear in garlic. In order to clarify the regulatory role of the ceRNA network in garlic somatic embryogenesis, we constructed lncRNA and miRNA libraries of four important stages (explant stage: EX; callus stage: AC; embryogenic callus stage: EC; globular embryo stage: GE) in the somatic embryogenesis of garlic. It was found that 44 lncRNAs could be used as precursors of 34 miRNAs, 1511 lncRNAs were predicted to be potential targets of 144 miRNAs, and 45 lncRNAs could be used as eTMs of 29 miRNAs. By constructing a ceRNA network with miRNA as the core, 144 miRNAs may bind to 1511 lncRNAs and 12,208 mRNAs. In the DE lncRNA-DE miRNA-DE mRNA network of adjacent stages of somatic embryo development (EX-VS-CA, CA-VS-EC, EC-VS-GE), by KEGG enrichment of adjacent stage DE mRNA, plant hormone signal transduction, butyric acid metabolism, and C5-branched dibasic acid metabolism were significantly enriched during somatic embryogenesis. Since plant hormones play an important role in somatic embryogenesis, further analysis of plant hormone signal transduction pathways revealed that the auxin pathway-related ceRNA network (lncRNAs-miR393s-*TIR*) may play a role in the whole stage of somatic embryogenesis. Further verification by RT-qPCR revealed that the lncRNA125175-miR393h-*TIR2* network plays a major role in the network and may affect the occurrence of somatic embryos by regulating the auxin signaling pathway and changing the sensitivity of cells to auxin. Our results lay the foundation for studying the role of the ceRNA network in the somatic embryogenesis of garlic.

## 1. Introduction

Garlic (*Allium sativum* L.) is an annual or biennial herb of the Allium genus in the Liliaceae family. Its product organ, the bulb, is unique in flavor, nutritious, economically and medicinally important, and popular worldwide [1,2]. However, most garlic varieties cannot form seeds, and the production of garlic relies on bulbs to reproduce asexually due to fertility degradation [3]. Infertility and long-term asexual reproduction of garlic have resulted in virus accumulation, species degradation, genetic improvement, and planting innovation difficulties [4], which not only seriously affect the yield and quality of garlic but also greatly restrict the breeding of new varieties of garlic.

Somatic embryo refers to the embryoid structure that originates from plant non-zygotic cells (somatic cells) and is formed through a process similar to embryogenesis and development [5]. Since the successful induction of somatic embryogenesis and regeneration of intact plants in carrot, somatic embryogenesis has been found in more than 1000 plants [6,7]. The regenerated plants formed by somatic embryogenesis have the advantages of fast speed, high reproduction coefficient, strong genetic stability, and neat regenerated individuals [8,9]. This can effectively solve the problems of virus accumulation, seed degradation, and germplasm innovation difficulties in garlic. Although somatic embryogenesis of garlic has been reported for a long time, it mainly focuses on the optimization of culture conditions, observation of morphological and anatomic characteristics, and analysis of physiological and biochemical mechanisms [10,11]. However, studies on gene expression and regulatory mechanisms in garlic during somatic embryogenesis are rarely reported.

Abundant studies have confirmed that microRNA (miRNA) plays an important role in plant somatic embryogenesis, and it has been reported in Arabidopsis thaliana [12], rice [13], citrus [14], longan [15], cotton [16], larch [17], and other species. During citrus somatic embryogenesis, miR156a/b and miR171c play a major role in the induction stage of embryogenic callus [14]. Studies on Arabidopsis showed that miR165/166 and miR160 promoted somatic embryogenesis by indirectly affecting the expression of LEC in explants and regulating auxin synthesis [12]. miR197 and miR528 play an important role in the maintenance of embryogenic callus in rice [13]. The above studies have shown that miRNAs play an important regulatory role in the process of plant somatic embryogenesis, and the regulatory effects at different stages have different characteristics. Although the role of miRNAs in plant somatic embryogenesis has been studied, there are few reports on the role of miRNAs in garlic somatic embryogenesis.

With the continuous development of high-throughput sequencing technology, abundant sequencing data have expanded the plant genome database, which has greatly accelerated the plant lncRNA research process. A number of lncRNA databases have been established for several species, such as tomato [18], apple [19], poplar [20], and pear [21], revealing important regulatory roles of lncRNA in plant flowering, seed germination, and response to biotic and abiotic stress. Based on RNA-Seq data from 35 different flower and fruit tissues of *Fragaria vesca* of diploid strawberry, Kang et al. (2015) found that a large number of lncRNAs were highly specifically expressed in mature pollen as compared with fruits [22]. In Arabidopsis thaliana, lnc351 was auxin-induced and competitively inhibited with the RNA-binding protein family (NSRs), preventing downstream genes from binding to NSRs and thereby affecting lateral root growth [23]. Li et al. (2018) found that lncRNA1459 was related to fruit ripening and affected ethylene production in tomato fruits. Knockout of lncRNA1459 by the CRISPR/Cas9 system will seriously delay the maturity of tomato [24]. Although there have been many reports on the roles of lncRNA in plant growth and development as well as plants’ response to biotic and abiotic stresses, there are few reports on lncRNA in plant somatic embryogenesis, especially for garlic.

Studies have shown that miRNA targets can produce indirect regulatory effects through the competitive binding of common microRNA reaction elements (MRE). Such regulatory patterns constitute competitive endogenous RNA (ceRNA) [25]. The ceRNA includes protein-coding RNA and non-coding RNA, such as lncRNA, pseudogene RNA, circRNA, etc. [26]. ceRNA was first discovered in Arabidopsis thaliana, and studies have shown that non-coding RNA IPS1 can affect PHO expression by binding to miR399 [27]. Subsequently, more and more studies have shown that ceRNA is widely present in plants and participates in various processes of plant growth and development [28,29]. However, to our knowledge, the ceRNA involved in the somatic embryogenesis of garlic has not been reported.

Therefore, based on the hypothesis that lncRNAs and mRNAs may play an important role in the somatic embryogenesis of garlic by competitively binding miRNAs, we isolated and identified lncRNAs, miRNAs, and mRNAs from four stages of garlic somatic embryogenesis by Illumina sequencing and bioinformatics analysis. The four stages included the explant stage (EX), callus stage (CA), embryogenic callus stage (EC), and globular embryo stage (GE). The functions of lncRNAs/miRNAs and their target genes (mRNAs) in these four stages were analyzed, and the possible regulatory effects of lncRNAs/miRNAs on somatic embryogenesis were investigated. Furthermore, the lncRNA-miRNA-mRNA regulatory network based on ceRNA at different stages of somatic embryogenesis was constructed, and its regulatory effects at different stages of garlic somatic embryogenesis were analyzed. The results enriched the ceRNA regulatory mechanism of garlic somatic embryogenesis and constructed ceRNA networks at different stages, which laid the foundation for revealing the potential functions of ceRNA and its regulatory network in garlic somatic embryogenesis.

## 2. Results

### 2.1. Morphological and Histological Characteristics of Garlic Somatic Embryos at Different Developmental Stages

The explants (EX) of the ‘Ershuizao’ garlic variety were inoculated on the embryonic callus induction medium (Figure 1A). When cultured for 13 days, the bottom of the explants expanded to form a faint-yellow substance, which was the induced callus (CA) (Figure 1D). The tissue sections showed that the callus cells were a group of small, irregularly shaped parenchyma cells, which were loosely arranged, highly vacuolated, with small or no obvious nuclei (Figure 1E,F). When cultured for 45 days, the callus formed obvious granular protrusions, which were embryonic callus (EC) (Figure 1G). The histological characteristics were small cells, closely arranged, and large and obvious nuclei (Figure 1H,I). With the prolongation of culture time, granular protrusions increased and the volume became larger, almost covering the original callus. When cultured for 90 days, the embryonic callus began to differentiate into semi-transparent, smooth spherical tissue, which was a globular embryo (GE) (Figure 1J). The globular embryo cells were small and regularly arranged with an obvious nucleus (Figure 1K,L).

The sampling stage of the current study is precise since it was based on the above morphological and histological observations together with our previous studies. Subsequent sequencing will be performed using the samples at specific stages.

### 2.2. Identification and Functional Analysis of lncRNA and mRNA during Somatic Embryogenesis of Garlic

#### 2.2.1. Identification and Characterization of lncRNA and mRNA during Somatic Embryogenesis of Garlic

In order to identify lncRNA during the somatic embryogenesis of garlic, the DNBSEQ platform was used to analyze the samples. There were 12 samples at the four stages (EX, CA, EC, and GE) of somatic embryogenesis, with three biological replicates per stage. Each sample produced an average of 111.02 M of raw data (Appendix A). After removing the redundant reads, an average of 106.57 M clean reads per sample was produced, with more than 85% of the clean readings matching the garlic full-length transcriptome. In order to obtain highly reliable garlic lncRNAs, candidate lncRNAs were identified by the following screening process: (Ⅰ) transcripts with length ≤200 bp and proteins with length >100 aa were removed; (Ⅱ) remove Nr annotation sequence; and (Ⅲ) CPC2, PLEK, CAPT, and SP were used to analyze the coding potential of transcripts.

After rigorous screening and identification, a total of 14,680 transcripts were identified as candidate lncRNAs (Figure 2A), among which 3694 lncRNAs were expressed in the four stages of somatic embryogenesis. Among the lncRNAs expressed only in the three stages of somatic embryogenesis, the number of lncRNAs expressed in the three stages of EX, CA, and EC was the largest (1304), while the number of lncRNAs expressed in the three stages of EX, CA, and GE was the least (108). Among lncRNAs expressed in only one stage, the number of EX stages was the largest (2451), while the number of CA stages was the smallest (382) (Figure 2B).

Further comparison of lncRNA and mRNA showed that the expression of lncRNA was lower than that of mRNA at all stages of somatic embryogenesis (Figure 2C). The length of lncRNAs ranged from 200 to 6064 bp, with an average length of 641 bp. Most lncRNAs (89.5%) were less than 1000 bp, and only 2.6% of lncRNAs were longer than 1500 bp. The length of mRNA transcripts ranged from 213 to 8947 bp, with an average length of 1471 bp (Figure 2D). The ORF length and GC content of lncRNA were lower than those of mRNA (Figure 2E,F).

#### 2.2.2. Differential Expression Analysis of lncRNA and mRNA during Somatic Embryogenesis of Garlic

In order to explore the potential functions of lncRNAs and mRNAs in garlic somatic embryogenesis, we first screened differentially expressed (DE) transcripts (FDR < 0.05, |log_2_FC| > 1) and constructed expression heat maps of 1246 DE lncRNAs and 7086 DE mRNAs at different stages of somatic embryogenesis (Appendix A). To confirm the reliability of those expression patterns, 16 highly expressed transcripts (including 8 lncRNAs and 8 mRNAs) were randomly selected for validation. Real-time quantitative PCR (RT-qPCR) was used to analyze the gene expression during somatic embryogenesis. Linear regression analysis showed a strong positive correlation (R^2^ = 0.8528 **), which indicated good reproducibility between transcript abundance assayed by RNA-seq and the expression profile revealed by RT-qPCR data (Appendix A). Overall, the results were highly consistent with the RNA-seq data.

In all differential transcripts, the number of DE mRNAs was significantly higher than that of DE lncRNAs at each stage (Figure 3A,B), and the number of DE mRNAs and DE lncRNAs gradually increased with the development of somatic embryos (Figure 3C,D). Comparing the differential transcripts at different stages of somatic embryogenesis, it was found that 94 mRNAs and 17 lncRNAs were differentially expressed at all stages of somatic embryogenesis, and these mRNAs and lncRNAs may play a role at all stages of somatic embryogenesis (Figure 3E,F).

#### 2.2.3. Function Prediction of Differentially Expressed lncRNA

To investigate the function of DE lncRNAs, RIblast was used to predict the trans-regulatory target genes of lncRNAs (Appendix A). Among the potential mRNA target genes of DE lncRNAs, there were 95 DE mRNA target genes in EX-VS-CA (37 up-regulated and 58 down-regulated), 168 DE mRNA target genes in CA-VS-EC (47 up-regulated and 121 down-regulated), 421 DE mRNA target genes in EC-VS-GE (136 up-regulated and 285 down-regulated) (Appendix A). In order to further clarify the functions of lncRNAs, Gene Ontology (GO) was used to annotate the DE target mRNA genes. The results showed that DE target mRNAs in different stages (EX-VS-CA, CA-VS-EC, and EC-VS-GE) were enriched in biological processes, cellular components, and molecular functions. In the biological process category, it is mainly concentrated in the metabolic process, biological regulation, and stress response (Figure 4). In the cell component category, it is mainly enriched in cells and cell components (Figure 4). In the molecular function category, it is mainly enriched in catalytic activity and binding (Figure 4).

The potential DE mRNA target genes were further classified using the Kyoto Encyclopedia of Genes and Genomes (KEGG) (Figure 5). It was found that 95 potential target genes were enriched in 37 signaling pathways in EX-VS-CA, among which 8 KEGG pathways such as zeatin biosynthesis, biosynthesis of secondary metabolites, and phenylpropane biosynthesis were significantly enriched. In CA-VS-EC, 168 potential target genes were enriched in 47 signaling pathways, among which 6 pathways such as zeatin biosynthesis, plant hormone signal transduction, and brassinosteroid biosynthesis were significantly enriched. In EC-VS-GE, 421 potential target genes were enriched in 80 signaling pathways, among which 14 signaling pathways, such as zeatin biosynthesis, MAPK signaling pathway-plant, and alpha-Linolenic acid metabolism, were significantly enriched. 

### 2.3. Identification and Functional Analysis of miRNA during Somatic Embryogenesis of Garlic 

In order to identify miRNAs during somatic embryogenesis and analyze their functions, we used BGISEQ-500 sequencing technology to obtain a total of 23,120,741–24,935,866 original sequences (Appendix A). After filtering out the low-quality sequences, a total of 22,007,205–23,832,648 target sequences were obtained. By comparing with the known small RNA database, it was found that the number of rRNA and miRNA was the largest, accounting for 47.69% and 45.39% of the total RNA, respectively (Figure 6A). By comparison with the miRBase database, 63 known miRNAs (belonging to 24 families) were identified. Characteristic analysis of these miRNAs revealed that the first bases of miRNAs with different lengths were biased towards U and A bases, and fewer contained C and G bases (Figure 6B). According to the feature that the precursor of miRNAs can fold to form a stable stem-loop structure, a total of 156 new miRNAs were predicted, with a length distribution of 19–30 bp. The miRNAs with 24 bp were the most abundant, and the first bases were biased toward U and A bases (Figure 6C). Expression analysis showed that 88 miRNAs were expressed in four stages. There were 7, 4, 1, and 5 miRNAs only expressed at the EX, CA, EC, and GE stages, respectively (Figure 6D).

By comparing DE microRNAs at different stages of somatic embryogenesis, 35 (21 up-regulated, 14 down-regulated), 20 (7 up-regulated, 13 down-regulated), and 26 (8 up-regulated, 18 down-regulated) DE miRNAs were found in EX-VS-CA, CA-VS-EC, and EC-VS-GE, respectively. Using psRNATarget to predict miRNA target genes (Expect ≤ 4), 146 miRNAs can target 10,278 mRNAs (Appendix A). Among them, in EX-VS-CA, 28 DE miRNAs targeted 46 DE mRNAs; in CA-VS-EC, 18 DE miRNAs targeted 44 DE mRNAs; and in EC-VS-GE, 24 DE miRNAs targeted 112 DE mRNAs.

KEGG analysis showed that 32 potential target genes were enriched in 14 pathways in EX-VS-CA, of which 3 pathways (α-linolenic acid metabolism, SNARE interaction in vesicle transport, amino sugar, and nucleoside sugar metabolism) were significantly enriched. In CA-VS-EC, 39 potential target genes were enriched in 19 signaling pathways, of which only the α-linolenic acid metabolism pathway was significantly enriched. In EC-VS-GE, 112 potential target genes were enriched in 44 signaling pathways, of which 5 signaling pathways, such as amino sugar and nucleotide metabolism, glyceride metabolism, metabolic pathways, α-linolenic acid metabolism, and plant hormone signal transduction, were significantly enriched (Appendix A).

### 2.4. lncRNA-miRNA Network Prediction

There are a large number of differentially expressed lncRNAs and miRNAs in the process of garlic somatic embryogenesis. To study the synergistic regulation of lncRNA–miRNA, analyze the relationship between lncRNA-miRNA in the process of garlic somatic embryogenesis and its potential role in garlic somatic embryogenesis. The analysis was carried out from different perspectives. They included lncRNA as a precursor of miRNAs, lncRNA as a target gene of miRNAs, and lncRNA as an endogenous target mimic (eTMs) of miRNAs.

The analysis revealed that 44 lncRNAs were highly homologous to 34 miRNA (including 26 novel miRNA) precursors (sequence alignment coverage greater than 90%) (Appendix A), indicating that these lncRNAs may act as precursors of miRNAs to regulate miRNA synthesis and transcription, thereby regulating the expression of downstream genes. Among them, lncRNA93304 is the precursor of miR156a/a-5p/j/k, and lncRNA184039 and lncRNA53284 are the precursors of miR160b. By annotating the predicted target genes of these miRNAs, it was found that the target genes of the miR156 family were involved in the transduction of brassinolide, and the target genes of miR160b were involved in the transduction of auxin. Thereby, lncRNA93304 (precursor of miR156a/a-5p/j/k) and lncRNA184039/lncRNA53284 (precursor of miR160b) may be involved in the regulation of these processes.

Some lncRNAs may also be regulated by miRNA targeting. We used the psRNATarget software (Expect ≤ 4) to predict possible miRNA targets of lncRNAs. A total of 1511 lncRNAs were predicted as 144 possible miRNA targets (Appendix A), and there were multiple interaction patterns with target genes. For example, the same lncRNA40568 is simultaneously regulated by three miRNA families; the miR156 family regulates 28 lncRNAs. Based on the above prediction, the lncRNA-miRNA regulatory network was constructed (Figure 7), where a large number of lncRNAs were regulated by miRNAs during garlic somatic embryogenesis.

In addition, lncRNA may also act as eTMs to bind to miRNA and block miRNA cleavage, thereby regulating miRNA function. Using the online software TAPIR, 45 lncRNAs were predicted to be eTMs of 29 miRNAs (8 known and 21 novel miRNA) (Appendix A). According to their sequence conservation, 8 known miRNAs come from 5 different miRNA families, miR159, miR160, miRNA396, miRNA399, and miR5139. Among them, three lncRNAs were identified as eTMs of miR159a (lncRNA64389) and miR160 (lncRNA184039, lncRNA53284), which were involved in the regulation of multiple hormone signaling pathways during plant growth and development. Therefore, the target genes of these 29 miRNAs were analyzed, and their functions were determined using KEGG analysis (Figure 8).

### 2.5. Construction of ceRNA Network in Adjacent Stages of Garlic Somatic Embryogenesis

Based on the theory that miRNA negatively regulates target RNA molecules, a ceRNA network with miRNA as the core and lncRNA and mRNA as targets was constructed. The 144 miRNAs were found to be associated with 1511 lncRNAs and 12,208 mRNAs. Considering that a graph cannot display all the network information between miRNA, lncRNA, and mRNA, primary miRNAs and their associated targets were selected to make the ceRNA network (Figure 9A). At the same time, the DE lncRNA-DE miRNA-DE mRNA regulatory network pattern map of the adjacent stages of somatic embryogenesis was constructed.

In the EX-VS-CA stage, 20 DE miRNAs were associated with 114 DE mRNAs and 18 DE lncRNAs (Figure 9B), including 216 pairs of lncRNA/mRNA-miRNAs. In the CA-VS-EC stage, 12 DE miRNAs were associated with 53 DE mRNAs and 12 DE lncRNAs (Figure 9C), including 104 pairs of lncRNA/mRNA-miRNAs. During the EC-VS-GE process, 14 DE miRNAs were associated with 88 DE mRNAs and 20 DE lncRNAs (Figure 9D), including 213 pairs of lncRNA/mRNA-miRNAs. By comparing the ceRNA networks at different stages, it was found that the ceRNA networks with miR156a\a-5p, miR164b, and miR393h as the core were differentially expressed in the adjacent stages of somatic embryogenesis, indicating that these three ceRNA networks may play a role in the whole process of somatic embryogenesis. In the ceRNA network with miR396 as the core, miR396f was differentially expressed in the EX-CA-EC stage, while miR396a and miR396e were differentially expressed in the EX-VS-CA and CA-VS-EC stages, respectively. The mRNA target genes of miR396a\e\f were all *GRF3/4*, and the lncRNA target gene was lncRNA107054. Therefore, lncRNA107054-miR396-*GRF3\4* may play a regulatory role in the EX-CA-EC process. In the network with miR399 as the core, miR399a\j was specifically highly expressed in the callus tissue stage, while the expression trend of its mRNA target gene and lncRNA target gene was opposite to that of miRNA. Therefore, the ceRNA network with miR399 as the core may play a regulatory role in the EX-CA-EC process. LncRNA208500-miR398b-*P450*\*GID* and lncRNA184039-miR160a\b\h-*SPATTULA* networks were only differentially expressed in the EX-VS-CA stage, which may play a role in callus induction. In the ceRNA network with miR394a as the core, the expression of miR394a decreased significantly in the EC-VS-GE stage, and the expression of its two mRNA target genes (*HIPL*) and two lncRNAs (lncRNA429326 and lncRNA242171) increased, indicating that the ceRNA network with miR394a as the core may play a regulatory role in the EC-VS-GE stage.

In order to reveal the potential role of lncRNA and miRNA in the ceRNA network, we performed GO and KEGG analysis of DE mRNA in adjacent stages (EX-VS-CA, CA-VS-EC, EC-VS-GE). GO analysis showed that DE mRNA in adjacent stages was enriched in biological processes, cellular components, and molecular functions (Appendix A). KEGG analysis showed that there were 8, 9, and 8 significant enrichment pathways in the EX-VS-CA, CA-VS-EC, and EC-VS-GE phases, respectively. Among them, plant hormone signal transduction, butanoate metabolism, and C5-branched dibasic acid metabolism were significantly enriched during somatic embryogenesis (Figure 9E–G).

Because plant hormones play an important role in somatic embryogenesis, we further studied plant hormone signal transduction pathways. It was found that hormone-related genes such as auxin, brassinosteroids, abscisic acid, and ethylene may play a role in somatic embryogenesis. It is worth noting that auxin-related genes include *TIR*, *ARF*, and *SAUR*; the *TIR*-related ceRNA network (lncRNAs-miR393a/h-*TIR*) may play a role in the whole stage of somatic embryogenesis, while the *ARF* and *SAUR*-related ceRNA network (lncRNA184039/lncRNA370646-miR160s-*ARF* and lncRNA115865/lncRNA182333-miR156g-*SAUR*) may play a role in the EX-VS-CA and CA-VS-EC stages, respectively. In addition, the brassinosteroids-related ceRNA network (lncRNA70477/lncRNA476489-miR156s-*SPB*) may play a role in the EX-VS-CA and CA-VS-EC stages; the abscisic acid-related ceRNA network (lncRNA208500/lncRNA73272-miR157d-*PP2C*) may play a role in the EX-VS-CA and EC-VS-GE stages; and the ethylene-related ceRNA network (lncRNA26786/lncRNA300029/lncRNA205817/lncRNA290148-miR167d-*ETR*) may play a role in the CA-VS-EC and EC-VS-GE stages.

### 2.6. Expression Identification of miR393s-ceRNA Network during Somatic Embryogenesis 

In order to study the possible role of the ceRNA network in somatic embryogenesis, we selected the miR393s-ceRNA network, which was differentially expressed in adjacent stages of somatic embryogenesis, for expression identification. During somatic embryogenesis, 6 lncRNA competed with 2 *TIR* (*TIR1* and *TIR2*) to bind to miR393h (sequencing identified four members of the miR393 family, but their precursor sequences were the same (Appendix A), and only miR393h was detected by stem-loop RCR) (Figure 10A). We first verified all lncRNAs by RT-qPCR and found that only lncRNA125175 had a high expression level, so we focused on it (Appendix A). It can be seen from Figure 10B that except for the CA stage, the expression of miR393h in the other somatic embryogenesis stages was basically the same as that of *TIR1*, but lower than that of *TIR2* and lncRNA125175. In addition, the expression trend of miR393h was opposite to that of *TIR2* and lncRNA125175, while the expression trend of miR393h was opposite to that of *TIR1* only in the EX-VS-CA phase and the same in the EC-VS-GE phase. These results suggest that the lncRNA125175-miR393h-*TIR2* network may play a major role in somatic embryogenesis.

## 3. Discussion

In vitro plant cell culture, plant regeneration by somatic embryogenesis is widely used [7]. As a special way of plant asexual reproduction, somatic embryogenesis is not only the research content and main method of cell development biology but also an important means of plant virus-free rapid propagation, germplasm resource preservation, and germplasm resource innovation [8,9]. There are many studies on the process and influencing factors of garlic somatic embryogenesis, as well as morphological disintegration and physiological and biochemical characteristics [10,11]. However, there is no research on lncRNA and miRNA in the process of garlic somatic embryogenesis, especially the molecular mechanism and regulatory mechanism of the lncRNA and miRNA-mediated regulatory network.

### 3.1. Expression Characteristics and Differences of lncRNA and miRNA in Different Stages of Garlic Somatic Embryogenesis

In recent years, significant progress has been made in the study of plant lncRNA, especially in model plants such as Arabidopsis thaliana and tomato, which are involved in the regulation of plant growth, flowering, and biotic and abiotic stress. However, the research on lncRNA involved in the regulation of plant somatic embryogenesis needs further investigation. Here 14,680 lncRNAs related to somatic embryogenesis were systematically identified in the four stages of garlic somatic embryogenesis (EX, CA, EC, and GE). In the four stages of garlic somatic embryogenesis, the expression of lncRNA in the same stage was lower than that of mRNA; there are 2451 lncRNAs specifically expressed in the EX stage, which is much higher than CA (382), EC (1402), and GE (1118). We found that lncRNA shows lower expression and stronger tissue specificity in garlic somatic embryogenesis as compared with protein-coding genes. Accordingly, this was potentially the main reason for the differences shown in *Brassica rapa* fertility [30], *Malus domestica* color [19], and *Solanum lycopersicum* ripening [18]. The expression of lncRNAs in specific stages showed that the number of differentially expressed lncRNA in EC-VS-GE stages was the highest (703), which was twice the number of differentially expressed lncRNA in EX-VS-CA and CA-VS-EC stages. Thus, in the process of garlic somatic embryogenesis, the formation of different stages requires the participation of a variety of lncRNAs that are responsible for different molecular mechanisms.

Previous studies have reported that somatic embryogenesis is regulated by many differentially expressed conserved and non-conserved miRNAs [31,32]. The miR156 family, a highly conserved microRNA family, is a negative regulator of SBP domain proteins [33]. High expression of miR156 can inhibit the SBP domain and accelerate somatic embryogenesis [13]. In our study, we found that the miR156 family continued to increase during somatic embryogenesis, suggesting that the miR156 family may be related to somatic embryogenesis. The NAC transcription factor family plays an important regulatory role in meristem formation and secondary growth [34]. MiR164 targets the NAC transcription factor family and regulates the occurrence of callus to embryogenic callus [35]. In our study, miR164b was significantly up-regulated during the EC-VS-GE stage, indicating the important role of miR164b in somatic embryogenesis. Meanwhile, we found that miR164b also targets UDP-glucuronide decarboxylase. In higher plants, UDP-glucose decarboxylase irreversibly catalyzes the biosynthesis of UDP-glucuronic acid into UDP-xylose, which is a wood-based donor for the synthesis of plant cell walls [36,37]. Thereby, the increased expression of miR164 during somatic embryogenesis leads to the down-regulation of the UDP-glucuronic acid decarboxylase gene and the NAC transcription factor. The UDP-glucuronic acid decarboxylase gene and NAC transcription factor may play an important role in the formation of the cell wall at various stages of somatic embryogenesis. In this study, the expression of miR157d, miR159a, miR396a/h, and miR398 increased gradually during somatic embryogenesis, which was consistent with the results in plant species [32]. The expression of miR394a and miR5139 gradually decreased during somatic embryogenesis, indicating that their target genes may play an important and positive role in somatic embryogenesis. The expression of the miR166 family was significantly downregulated during callus induction in our study, which could be due to dedifferentiation stimuli [38].

### 3.2. Correlation and Characteristics of miRNA with lncRNA and mRNA in Different Stages of Garlic Somatic Embryo Development

LncRNA can be used as a precursor of miRNA and play a regulatory role by producing miRNA. Among the 14,680 lncRNAs, 45 lncRNAs can be used as precursors of miRNA, mainly miR156a/a-5p/j/k, miR160b, etc. There are multiple lncRNAs that act as precursors of a miRNA, such as lncRNA184039 and lncRNA53294 for miR160b; one lncRNA acts as a precursor to multiple miRNAs, such as lncRNA93304, a precursor to the miR156 family. MiR156 is closely related to somatic embryogenesis (Luo et al., 2006); in longan, miR160 regulates somatic embryogenesis by cleaving ARF10/16/17 in response to 2,4-D and ABA signals [15]. Therefore, lncRNAs may act as precursors of miRNAs to regulate the occurrence of garlic somatic embryos.

LncRNA can also be used as a target gene of miRNA to regulate plant growth, flowering, and fruit ripening at the post-transcriptional level [39]. In this study, 1511 lncRNAs were predicted to be the targets of 144 miRNAs by psRNAtarget. Among them, miR160a/b/h can target lcnRNA312348, lcnRNA184039, and lcnRNA53284; at the same time, lcnRNA184039 can be used as a precursor gene for miR160b. The MiR160 family has been reported to regulate somatic embryogenesis by cleaving ARF genes (mRNA target genes) [15]. However, the complex regulatory network of LncRNAs (lcnRNA312348, lcnRNA184039, and lcnRN53284) competing with *ARF* to bind miR160s and lcnRNA184039 as a precursor of miR160b has not been reported, which needs further investigation.

In addition, lncRNA can also be used as an endogenous target mimic for miRNA to bind to miRNA, thereby promoting the expression of mRNA targets. In Arabidopsis, lncRNA IPS1 binds to miR399 and forms three nucleotide bumps between bases 10 and 11 at the 5’ end of the miRNA, thereby inhibiting the cleavage of its target *PHO2* by miR399 [27]. In cotton, lnc_973 and lnc_253 can respond to salt stress as endogenous target mimics for miR399 and miR156 [29]. In our study, several eTMs modules such as miR168a-lncRN409438 and miR5139-lncRNA134026/lncRNA199633/lncRNA443538 were predicted. MiR168 is a highly conserved miRNA family that mainly targets AGO genes and ARF transcription factors. Chen et al. (2021) found that *DlAGOs* regulated early and middle somatic embryogenesis in longan [40]. Wójcikowska et al. (2017) found that 14 ARF transcription factors were involved in somatic embryogenesis in auxin-induced Arabidopsis embryo cultures [12]. Therefore, lncRNA409438 may participate in garlic somatic embryogenesis through the ceRNA network of miR168a.

miRNA plays an important role in plant life through the direct cutting of target mRNA or translation inhibition mediating post-transcriptional regulation. In our study, the potential DE mRNA target genes of DE miRNAs in EX-VS-CA, CA-VS-EC, and EC-VS-GE were predicted, and the DE mRNA was enriched by KEGG. It was found that the α-linolenic acid metabolic pathway was significantly enriched in these three processes. Somatic embryogenesis is a dynamic process involving changes in the contents of proteins, sugars, fatty acids, nucleic acids, free amino acids, and other substances [41]. α-linolenic acid is a kind of polyene unsaturated fatty acid, which is the basic component of the cell membrane [42] and can initiate jasmonic acid biosynthesis. Its endogenous content is directly related to the amount of jasmonic acid production [43]. Therefore, there may be a regulatory network of miRNA-αlinolenic acid-cell membrane/jasmonic acid during garlic somatic embryogenesis.

### 3.3. Regulatory Network and Potential Functions of ceRNA during Somatic Embryogenesis in Garlic

The functional mechanism of competitive endogenous RNA (ceRNA) has been widely used to study the role of miRNA in plants. LncRNA can be used as eTMs to block the interaction between miRNAs and their mRNA target genes by simulating target genes as targets of specific miRNAs. The lncRNA23468 promotes tomato resistance to late blight by inhibiting miR482b to increase the expression level of NBS-LRR [44]. As an eTM of miR160b, cotton lncRNA354 regulates auxin response factors that play a role in salt stress tolerance [45]. In our study, DE lncRNA-DE miRNA-DE mRNA networks were constructed in EX-VS-CA, CA-VS-EC, and EC-VS-GE stages, respectively, and KEGG enrichment of DE mRNA in each stage was further performed. It was found that plant hormone signal transduction, butyric acid metabolism, and C5 branched-chain dicarboxylic acid metabolism were significantly enriched in each stage.

In the process of plant somatic embryogenesis, hormones play a vital role, although the dosage is small. Among the hormone signal transduction pathways we enriched, the auxin pathways-related genes and the ceRNA network are involved in the whole somatic embryogenesis process. The ceRNA network with miRNA393h as the core was differentially expressed in the adjacent stages of somatic embryogenesis. RT-qPCR showed that the expression trend of miR393h was opposite to that of its target genes, lncRNA125175 and *TIR2*, indicating that there was an interaction between the three transcripts that regulated somatic embryogenesis. In addition, the ceRNA network associated with *ARF* and *SAUR* (lncRNA184039/lncRNA370646-miR160s-*ARF* and lncRNA115865/lncRNA182333-miR156g-*SAUR*) may function in the EX-VS-CA and CA-VS-EC phases. Although there are many reports about the regulation of somatic embryogenesis by *ARF*, *SAUR,* and *TIR*, the interaction among them and the interaction among their networks have not been reported. Although there are many reports on the regulation of somatic embryogenesis by *ARF*, *SAUR,* and *TIR*, the interaction among them and the interaction among the networks in which they participate have not been reported. Therefore, it is of great significance to study the interaction between the networks involved in *ARF*, *SAUR,* and *TIR* and to regulate the development of somatic embryos.

## 4. Materials and Methods

### 4.1. Plant Material and Culture

The test material is the early maturing garlic variety “Ershuizao”, which is preserved in the College of Horticulture, Nanjing Agricultural University. The dormant garlic bulbs were selected, and the outer bract leaves were removed after sterilization. Afterwards, the storage leaves and nutrient leaves were removed, leaving about a 0.5 cm-thick bulb disc. Each bulb disc with a shoot tip was cut into 4 parts and inoculated on callus induction medium (B5 + 2,4-D 2 mg/L + KT 0.5 mg/L + sucrose 30 g/L + agar 6.5 g/L, pH 5.8), and cultured in the dark at 25 °C. Different stages of garlic somatic embryogenesis were determined by stereomicroscope and paraffin section (explants stage: EX; callus stage: AC; embryogenic callus stage: EC; globular embryo stage: GE). The materials at different stages were stored at −80 °C for later use.

### 4.2. Total RNA Extraction

The ethanol precipitation protocol and CTAB-PBIOZOL reagent were used for the purification of total RNA from the plant tissue according to the manual instructions. Grind tissue samples of about 80 mg with liquid nitrogen into powder and transfer the powdered samples into 1.5 mL of preheated 65 °C CTAB-pBIOZOL reagents. The samples were incubated by a thermomixer for 15 min at 65 °C to permit the complete dissociation of nucleoprotein complexes. After centrifuging at 12,000× *g* for 5 min at 4 °C, the supernatant was added to 400 µL of chloroform per 1.5 mL of CTAB-pBIOZOL reagent and centrifuged at 12,000× *g* for 10 min at 4 °C. The supernatant was transferred to a new 2.0 mL tube that contained 700 µL acidic phenol and 200 μL chloroform, followed by centrifugation at 12,000× *g* for 10 min at 4 °C. The aqueous phase was added at an equal volume to the aqueous phase of chloroform and centrifuged at 12,000× *g* for 10 min at 4 °C. The supernatant was added in equal parts to the isopropyl alcohol and left at −20 °C for 2 h to precipitate. After that, the mix was centrifuged at 12,000× *g* for 20 min at 4 °C and then the supernatant was removed. After being washed with 1 mL of 75% ethanol, the RNA pellet was air-dried in the biosafety cabinet and dissolved by adding 50 µL of DEPC-treated water. Subsequently, total RNA was qualified and quantified using a Nano Drop and an Agilent 2100 bioanalyzer (Thermo Fisher Scientific, Waltham, MA, USA).

### 4.3. RNA Library Construction

The non-coding RNA library was constructed as follows. Approximately 1 µg total RNA per sample was treated with the Ribo-Zero™ Magnetic Kit (Plant Leaf) (Epicentre) to deplete rRNA. The retrieved RNA is fragmented by adding First-Strand Master Mix (Invitrogen). First-strand cDNA was generated using random primers and reverse transcription, followed by second-strand cDNA synthesis. The synthesized cDNA was subjected to end-repair and then was 3′ adenylated. Adapters were ligated to the ends of these 3′-adenylated cDNA fragments. Several rounds of PCR amplification with PCR Primer Cocktail and PCR Master Mix are performed to enrich the cDNA fragments. Then, the PCR products are purified with Ampure XP beads. The quality of the final library was examined by checking the distribution of the fragment sizes using the Agilent 2100 bioanalyzer. The quantity of the library was checked using real-time quantitative PCR (RT-qPCR) (TaqMan Probe). The qualified libraries were sequenced pair-end on the DNBSEQ platform (BGI-Shenzhen, Shenzhen, China).

The miRNA library was constructed in the following steps. The library was prepared with 1 µg total RNA for each sample. Total RNA was purified by electrophoretic separation on a 15% urea denaturing polyacrylamide gel electrophoresis (PAGE) gel, and small RNA regions corresponding to the 18–30 nt bands in the marker lane (14–30 ssRNA ladder marker, TAKARA) were excised and recovered. Afterwards, the 18–30 nt small RNAs were ligated to adenylated 3′ adapters annealed to unique molecular identifiers (UMI), followed by the ligation of 5′ adapters. The adapter-ligated small RNAs were subsequently transcribed into cDNA by superscript II reverse transcriptase (Invitrogen, Waltham, MA, USA), and then several rounds of PCR amplification with PCR Primer Cocktail and PCR Mix were performed to enrich the cDNA fragments. The PCR products were selected by agarose gel electrophoresis with target fragments 110~130 bp and then purified by the QIAquick Gel Extraction Kit (QIAGEN, Valencia, CA, USA). The quality and quantity of the miRNA library were detected in the same way as those of the non-coding RNA library. The final ligation PCR products were sequenced using the BGISEQ-500 platform (BGI-Shenzhen, Shenzhen, China).

### 4.4. Reads Mapping and Identification of lncRNA

The sequencing data were filtered with SOAPnuke (v1.5.2). The reads with sequencing adapter, low-quality base ratio (base quality less than or equal to 5) more than 20% or unknown base (‘N’ base) ratio of more than 5% were removed. Afterward, clean reads were obtained and stored in FASTQ format. The clean reads were mapped to the full transcriptome using HISAT2 (v2.0.4). After that, Ericscript (v0.5.5) and rMATS (V3.2.5) were used to detect fusion genes and differential splicing genes (DSGs), respectively. Bowtie2 (v2.2.5), a database built by BGI (Beijing Genomic Institute in Shenzhen), was applied to align the clean reads to the gene set. The known and novel, coding and noncoding transcripts were detected. Then the expression level of a gene was calculated by RSEM (v1.2.12). To identify lncRNAs with high confidence, the obtained transcripts were screened using the following criteria. Firstly, sequences with transcript length ≤200 bp and protein length >100 aa were removed. Secondly, the Nr annotation sequence was removed. Thirdly, the filtered transcripts per million fragments per million bases (FPKM) were smaller than 0.5. The last step was that Coding Protein Calculator2 (CPC2), a predictor of long non-coding RNAs and messenger RNAs based on an improved K-mer scheme (PLEK), Coding Potential Assessment Tool (CAPT), and Swiss Prot Database (SP), were used to remove transcripts with coding ability ≥0. The remaining transcripts are lncRNAs. Using DESeq2 (v1.4.5) to analyze differentially expressed genes, FDR ≤ 0.05 (false discovery rate) and |log_2_FC| > 1 were threshold values.

The raw sequencing data are called raw tags, which were processed using the following criteria. Low-quality tags, tags with 5 primer contaminants or poly A, tags without 3 primers or insertions, and tags shorter than 18 nt were removed. After filtering, the clean tags were mapped to the full transcriptome and other sRNA databases, including miRBase, siRNA, and snoRNA, with Bowtie2. Particularly, cmsearch was performed for Rfam mapping. The software miRNA was used to predict novel miRNAs by exploring the secondary structure. The small RNA expression level was calculated by counting absolute numbers of molecules using unique molecular identifiers. Differential expression analysis was performed using the DEGseq, where Q value ≤ 0.001, and the absolute value of Log_2_Ratio ≥ 1 as the default threshold to judge the significance of expression difference.

### 4.5. Target Gene Prediction and Functional Enrichment Analysis

Potential mRNA target genes of lncRNAs were predicted by RIblast software (e ≤ −30) [46]. The miRNAs’ target genes were predicted using the PsRNATarget online website (www.zhaolab.org/psRNAtarget) (accessed on 4 June 2022) (e ≤ 4) [47]. The differentially expressed mRNA target genes were analyzed by OmicShare (www.omicshare.com/tools) (accessed on 23 May 2022) for GO and KEGG enrichment, and q values were used to screen the significantly enriched pathways.

### 4.6. Prediction of the Relationship between lncRNA and miRNA

Some lncRNAs can form miRNA precursors through intracellular cleavage. In order to predict whether lncRNA is a miRNA precursor, blast software was used to compare lncRNA and miRNA precursors (lncRNA with a matching rate of ≥90% was selected as an miRNA precursor). In order to predict whether lncRNA is the target gene of miRNA, lncRNA and miRNA data were submitted to psRNATarget (Expect ≤ 4) to predict the target gene of miRNA. TAPIR through an online website (www.bioinformatics.psb.ugent.be/webtools/tapir/) (accessed on 6 June 2022) predicted whether lncRNAs trap targets for miRNAs endogenous analog (eTMs). Cytoscape (v3.5.1) software was used to generate network diagrams of the relationships between lncRNAs and related microRNAs and mRNAs.

### 4.7. Expression Analysis by RT-qPCR

The expression levels of mRNA and lncRNA in each sample were determined using the PowerUp^TM^ SYBRTM Green Master Mix kit. The miRNA expression level was concurrently determined in each sample by stem-loop RT-qPCR. *AsBES* was used as the internal reference gene. The relative expression level of genes was calculated using the Change 2^−ΔΔCt^ method.

The primers required for the quantification of mRNA, lncRNA, and miRNA are shown in Appendix A.

### 4.8. Data Statistics and Analysis

All the measurements contained three biological replicates. An analysis of variance (ANOVA) was performed using SPSS 26.0 at the *p* < 0.05 level. Origin 2023 was applied to make the graphs.

## 5. Conclusions

In this study, a total of 14,680 lncRNAs and 219 miRNAs were identified in the four stages of somatic embryogenesis (EX, CA, EC, and GE). The 44 lncRNAs were predicted to be used as precursors for 34 miRNAs, while 1511 lncRNAs could be the targets of 144 miRNAs. The 45 lncRNAs were predicted to be the eTMs of 29 miRNAs. Several lncRNA-miRNA-mRNA regulatory networks were constructed during somatic embryogenesis, among which the ceRNA network with miR393h as the core was differentially expressed in the adjacent stages of somatic embryogenesis in garlic. RT-qPCR showed that lncRNA125175-miR393h-*TIR2* plays a major role in the network and may affect the occurrence of somatic embryos by regulating the auxin signaling pathway and changing the sensitivity of cells to auxin. Our results contribute to understanding the regulatory role of ceRNA networks in the somatic embryogenesis of plants.

## Figures and Tables

**Figure 1 ijms-24-05311-f001:**
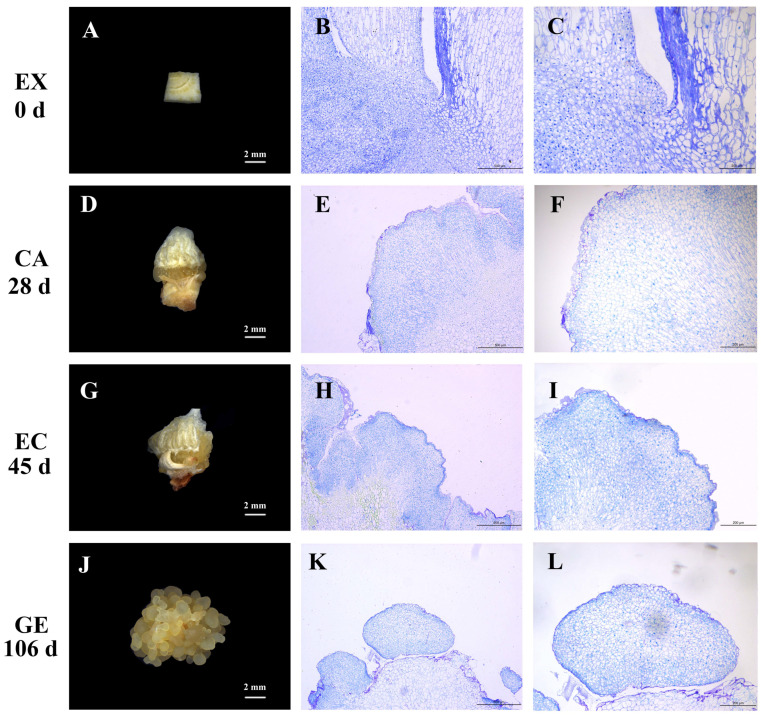
Morphological and histological observation at different stages of somatic embryo induction in garlic. Explants (**A**); paraffin sections of explants (**B**,**C**); callus (**D**); paraffin sections of callus (**E**,**F**); embryogenic callus (**G**); paraffin sections of embryogenic callus (**H**,**I**); globular embryo (**J**); paraffin sections of globular embryos (**K**,**L**). EX, CA, EC, and GE on the left side indicated different sampling stages.

**Figure 2 ijms-24-05311-f002:**
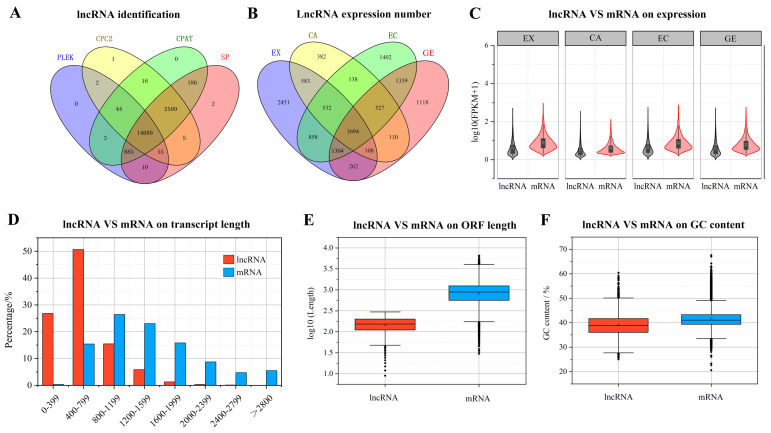
Screening, identification, and expression of lncRNA during somatic embryogenesis of garlic. The coding potential of lncRNA was analyzed by CPC2, PLEK, CPAT and SP (**A**); comparison of lncRNA specificity at different stages of somatic embryogenesis (**B**); lncRNA and mRNA expression abundance comparison (**C**); lncRNA and mRNA transcript length comparison (**D**); lncRNA and mRNA ORF length comparison (**E**); comparison of lncRNA and mRNA GC content (**F**).

**Figure 3 ijms-24-05311-f003:**
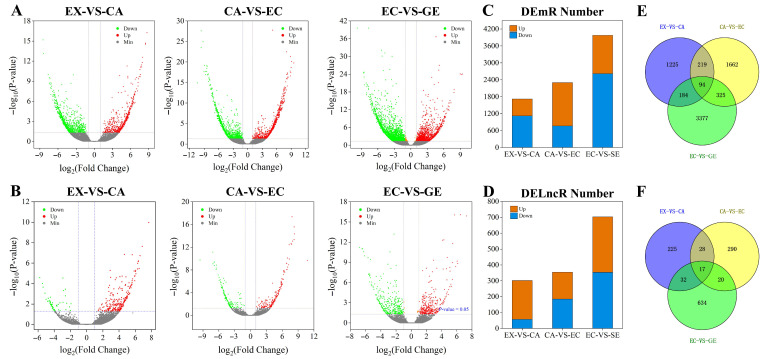
Differential expression analysis of lncRNAs and mRNAs during somatic embryogenesis of garlic. The differential expression volcano maps of mRNAs and lncRNAs at different stages (**A**,**B**); the number of differentially expressed mRNAs and lncRNAs at different stages of somatic embryo development (**C**,**D**). Orange was up-regulated and blue was down-regulated. Venn diagrams of differential expression of mRNAs and lncRNAs at different stages (**E**,**F**).

**Figure 4 ijms-24-05311-f004:**
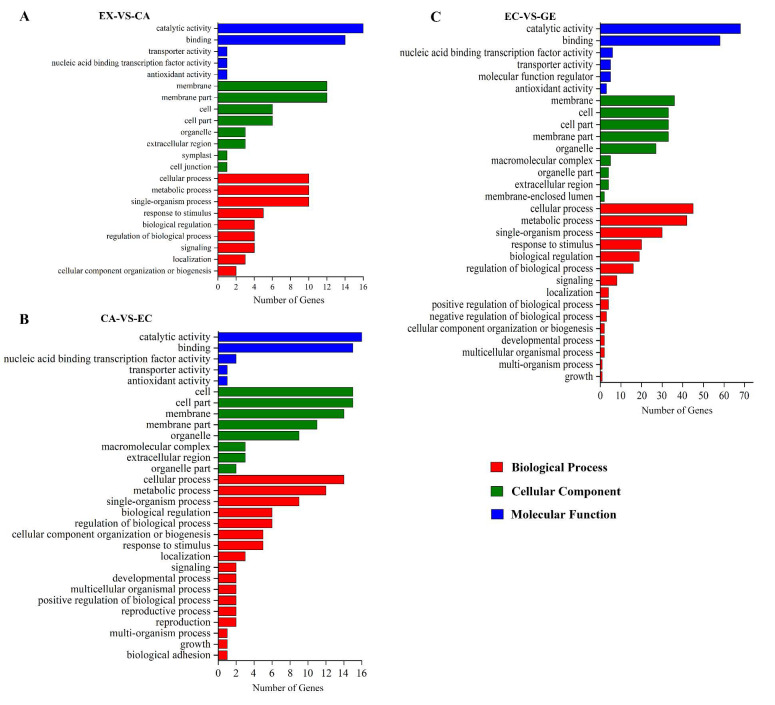
Gene Ontology analysis of potential DE mRNA target genes of lncRNA. (**A**–**C**) were gene ontology analyses of potential DE mRNA target genes of lncRNA in EX-VS-CA, CA-VS-EC, and EC-VS-GE stages, respectively; red bars indicated a biological process; green bars indicated the cell component; blue bars indicated the molecular function.

**Figure 5 ijms-24-05311-f005:**
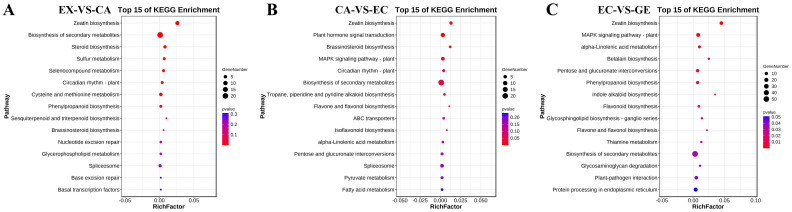
KEGG analysis of potential DE mRNA target genes of lncRNA. (**A**–**C**) represent the KEGG enrichment pathway maps of DE mRNA target genes of DE lncRNA in EX-VS-CA, CA-VS-EC, and EC-VS-GE stages, respectively.

**Figure 6 ijms-24-05311-f006:**
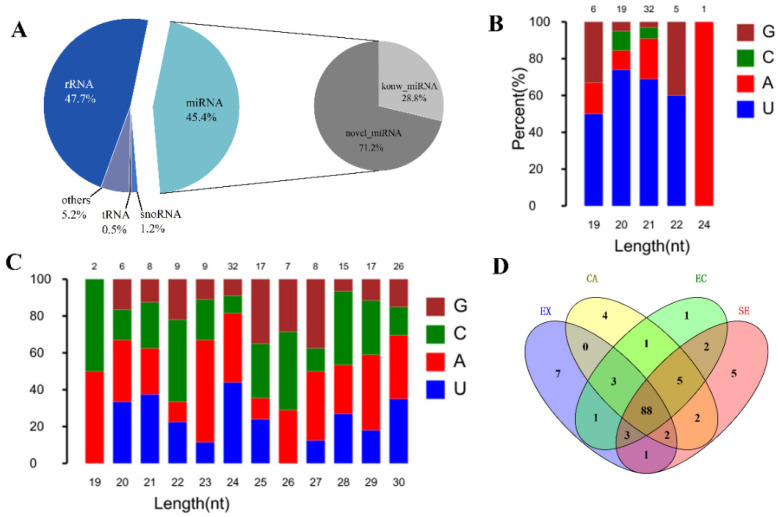
Screening, identification, and expression analysis of miRNA during garlic somatic embryogenesis. Small RNA classification and proportion (left), known and newly identified microRNA proportion (right) (**A**); the first base bias map of known miRNAs and newly identified miRNAs (**B**,**C**); comparison of specific expression of miRNA in different stages of somatic embryogenesis (**D**).

**Figure 7 ijms-24-05311-f007:**
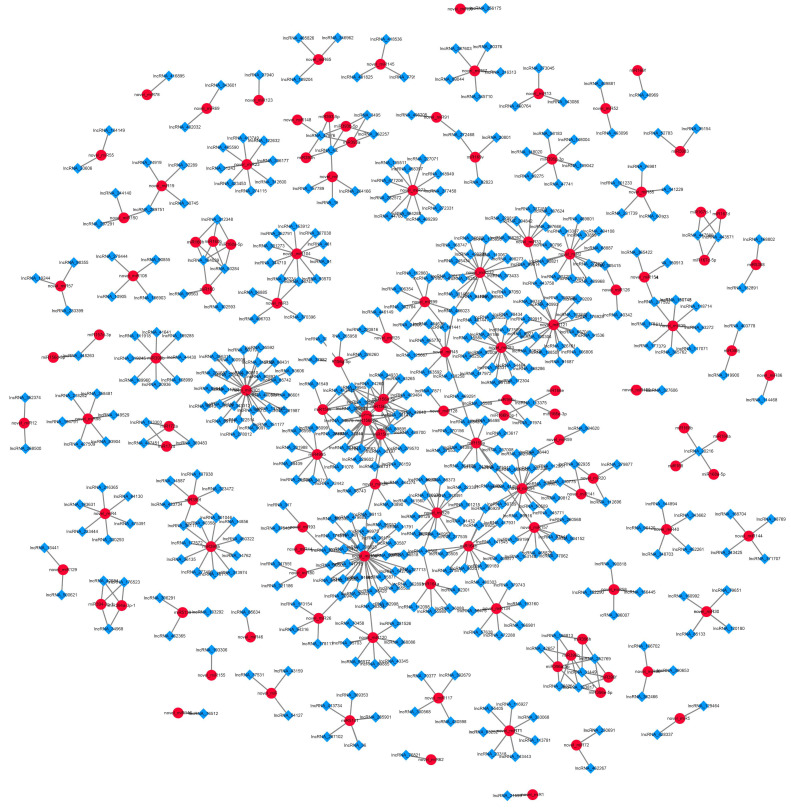
Prediction of lncRNA-miRNA relationship during garlic somatic embryogenesis.

**Figure 8 ijms-24-05311-f008:**
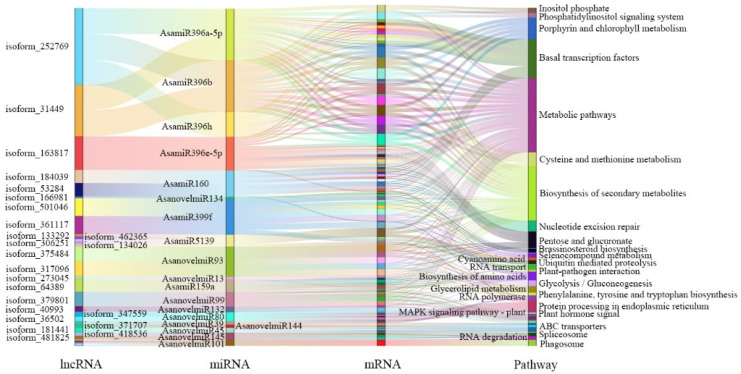
Sankey diagram of eTM-KEGG network in garlic somatic embryogenesis.

**Figure 9 ijms-24-05311-f009:**
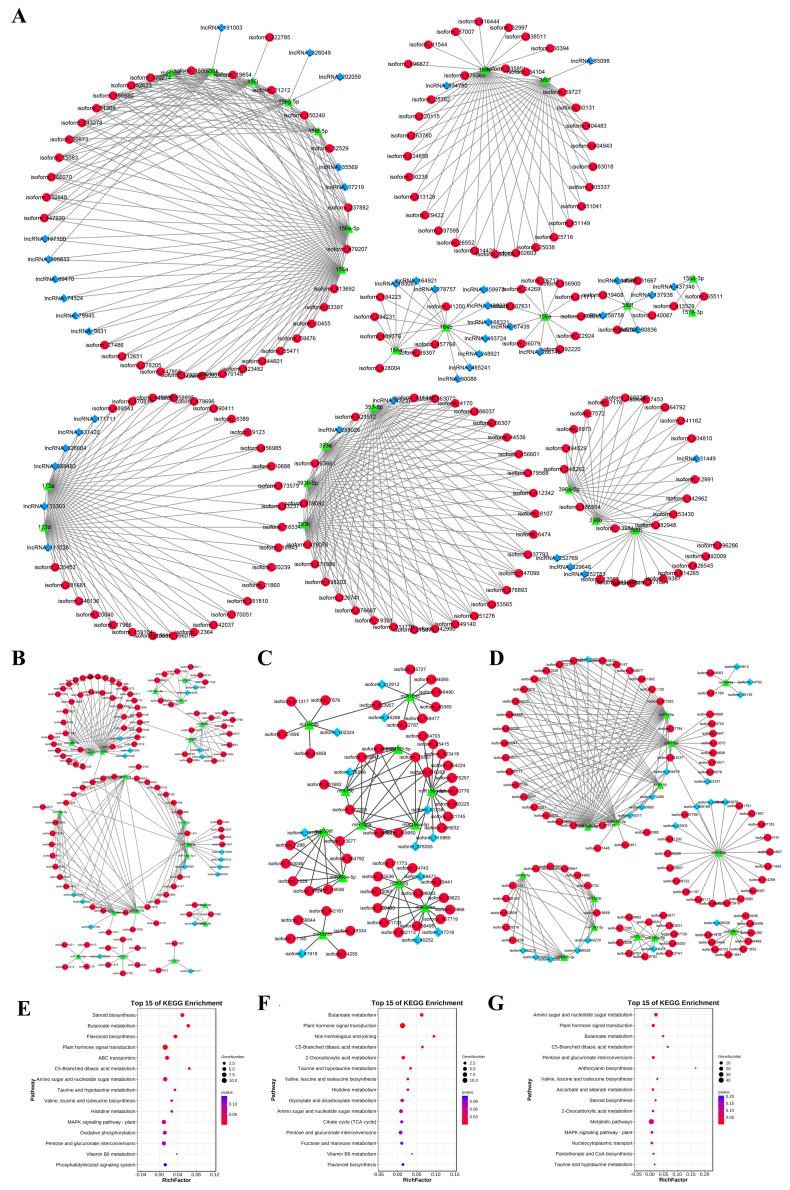
Construction of ceRNA network during garlic somatic embryogenesis. ceRNA Network during Somatic embryogenesis (some important nodes) (**A**); DE lncRNA–DE miRNA–DE mRNA networks of EX-VS-CA, CA-VS-EC, and EC-VS-GE stages, respectively (**B**–**D**); KEGG analysis of DE mRNA at EX-VS-CA, CA-VS-EC, and EC-VS-GE stages, respectively (**E**–**G**). Note: the blue diamond is lncRNA; the red circle is mRNA; the green triangle is miRNA.

**Figure 10 ijms-24-05311-f010:**
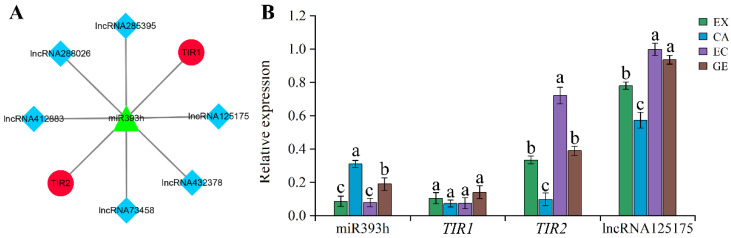
miR393s-ceRNA network and its related genes were verified by RT-qPCR. lncRNAs-miR393s-TIR network (**A**); miR393h-ceRNA network RT-qPCR validation (**B**). Lowercase letters indicate a significant difference at the 0.05 level (ANOVA analysis).

## Data Availability

All data generated or analyzed during this study are included in this published article.

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
