# Peer review of "Construction of ceRNA Networks at Different Stages of Somatic Embryogenesis in Garlic"

_ijms, 2023, doi:10.3390/ijms24065311_

Round 1

Reviewer 1 Report

The present manuscript, entitled Construction of ceRNA networks at different stages of somatic embryogenesis in garlic presented a very impressive and high-quality set of results. It was a pleasure to have read this work and the authors showed a very relevant strategy for understanding somatic embryogenesis and also presented very concise results. I was delighted reading the paper. Congratulations!

Reviewer 2 Report

The work described by Bai et al. is very interesting.

However, the Introduction part is too much lengthy, please reduce it to some extent.

Line 38-39: Reorganize this sentence

Line 145: change identification to “Identification”

Reviewer 3 Report

1. The authors should summarize the general control mechanism of somatic embryogenesis (SE) from the literature and then connect key genes of each stage to the part of the downstream ceRNA network. Importantly, the authors need to explore control mechanisms of phase transitions or critical events of SE, such as the initiation of proembryo, globular embryo, heart embryo, shoot apical meristem initiation, etc., and their connections to ceRNA networks.

2. The authors need to compare the difference in ceRNA networks between embryogenic and non-embryogenic calluses, which can find out the controlling factors of the capacity of embryogenesis from the callus.

3. The histological evidence for stages of SE looks not satisfactory, e.g. there is a lack of highlighting from non-embryogenic callus and the difference between embryogenic callus, also, the bi-polar structure of SE should be provided to prove a successful event of SE.

4. Add the results of significant effects between means to Figure 10B.

5. Add a paragraph of statistics to the section of M&M. How many replicates were conducted in each treatment?

6. L31-33, L629-630: It needs to describe more clearly the major role of the lncRNA125175-miR393h-TIR2 network, and give future direction more specific. Maybe organizing a conclusive graph will be helpful.

7. Figure 1: The timing for each stage of SE needs to be documented.

8. Overall, the study did not get any significant progress on the developmental biology of SE, particularly on the control mechanism involving ceRNA networks for any of the events, including phase transition, SAM initiation, embryogenic capacity acquisition, organ formation, etc.

Round 2

Reviewer 3 Report

I don't have further questions.